# DistDD: Distributed Data Distillation Aggregation through Gradient Matching

## Abstract

In this paper, we introduce DistDD, a novel approach within the federated learning framework that reduces the need for repetitive communication by distilling data directly on clients' devices. Unlike traditional federated learning that requires iterative model updates across nodes, DistDD facilitates a one-time distillation process that extracts a global distilled dataset, maintaining the privacy standards of federated learning while significantly cutting down communication costs. By leveraging the DistDD's distilled dataset, the developers of the FL can achieve just-in-time parameter tuning and neural architecture search over FL without repeating the whole FL process multiple times. We provide a detailed convergence proof of the DistDD algorithm, reinforcing its mathematical stability and reliability for practical applications. Our experiments demonstrate the effectiveness and robustness of DistDD, particularly in non-i.i.d. and mislabeled data scenarios, showcasing its potential to handle complex real-world data challenges distinctively from conventional federated learning methods. We also evaluate DistDD's application in the use case and prove its effectiveness and communication savings in the NAS use case.

## 1 Introduction

Federated learning typically involves iterative communication between the central server and its clients. Throughout the training process, the server proposes parameters for the clients to calculate the updates for their local models Zhou et al. (2021); Khodak et al. (2020); Agrawal et al. (2021). The server then aggregates these updates to refine the global model. While these communication costs might be necessary for the federated learning paradigm to maintain users' privacy, they become significant because a good machine learning model typically requires repeated training to debug better parameters and neural network architectures Zhang et al. (2021).

For example, consider the following two use cases:

**Use case A (Parameter Tuning)**. (see Figure 1): Considering the developers need to tune the hyper-parameters of the FL process, Khan et al. (2023); Zhang et al. (2021); Zhou et al. (2021); Agrawal et al. (2021) such as batch size, learning rate, epoch, optimizer, etc. In a typical FL architecture, the parameter tuning process requires repeating the full FL process, which involves multiple clients joining. Such a process brings enormous communication costs due to the unnecessary multiple repeat tuning.

**Use case B (NAS over FL).** (see Figure 1): Another example is the neural architecture search over FL Zhu et al. (2021); Zhu & Jin (2021); Liu et al. (2023a); He et al. (2021); Khan et al. (2023); Yan et al. (2024). Considering the scene in which the developers of the FL want to search for the optimal neural architecture for the FL tasks, The FL server must search for a new neural architecture at each iteration during such a process. Then, the FL server needs to perform the whole FL process using the searched architecture to collect the performance as feedback. Such approaches must be repeated multiple times until the optimal neural architecture is searched. This process brings huge communication costs as well.

Such use cases require repeatedly tuning the model, bringing huge communication costs Zhou et al. (2021). To reduce such communication costs, an appealing approach is for the clients to upload the data directly to

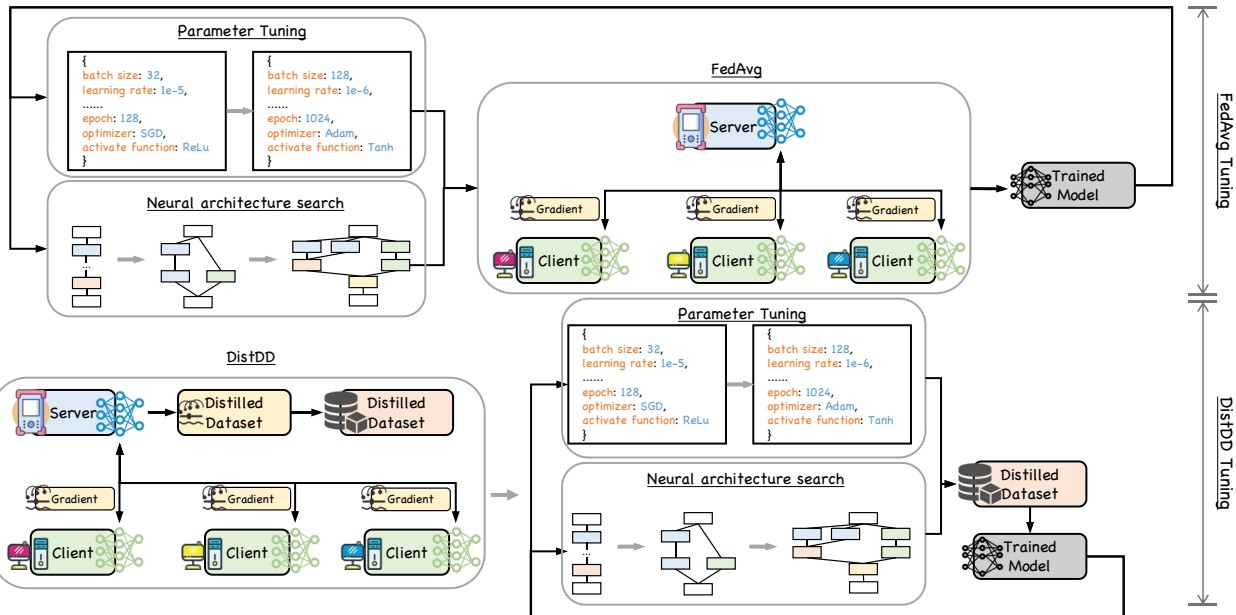

Figure 1: We provided two use cases for DISTDD. The parameter tuning and NAS require multiple whole FL processes for typical FL. For DISTDD, the server needs to acquire the distilled dataset through DISTDD process at first, then repeat local tuning and NAS within the FL server itself to get the optimal network architecture and optimal parameter. Then, the server can repeat the FL process only once using the optimal network architecture and optimal parameters.

the server so that future training and tuning can only happen within the server. However, an obvious flaw is that data uploading will invade the clients' privacy, which is against the principle of federated learning.

Therefore, in this paper, we seek to answer the question: How can we allow clients to upload the essential information to train a classifier so that the server can further train and tune the models without additional communication costs while protecting the client's privacy (as much as federated learning can protect).

To answer this question, we introduce a novel distributed data distillation method (Distributed Data Distillation through gradient matching) in this paper. DISTDD is a method that combines gradient matching Zhao et al. (2020) with distributed learning to distill knowledge from multiple clients into a single dataset. In this process, clients use their local datasets to get gradients. The critical step is to compute the loss between the aggregated global gradient and the gradient from the distilled dataset and use this loss to build the distilled dataset. Finally, the server uses the synthesis distilled dataset to tune and update the global model.

We demonstrate the convergence of DISTDD by conducting detailed experiments (§4.2). Since the non-iid and the mislabeling problems are frequently met in the optimization process of standard FL settings, we evaluated DISTDD under the non-iid (§4.4) and mislabeling problems (§4.3). These experimental results verify the effectiveness and robustness of DISTDD in dealing with complex real-world situations. We designed rigorous experiments and tested them on multiple public datasets. Furthermore, we also conducted a detailed ablation study (§C) and evaluated DISTDD's performance in the neural architecture use case (§4.5).

Overall, DISTDD provides a new approach for solving challenges in the current distributed big data environment and opens up new possibilities for future research. We believe that DISTDD will trigger more research in the fields of distributed learning and data distillation in the future. We summarize our contribution as three-folded:

- We propose a new method called DISTDD, which distillates data from distributed clients in a distributed way. DISTDD effectively distills the distributed data from the distributed clients, thus enabling numerous debugging attempts without more communication cost.

- We identify the potential mislabeling and non-iid problems in the distributed data distillation paradigm and propose methods to solve them.
- We provide convergence proof and conduct extensive experiments to prove the effectiveness and convergence of our DISTDD framework.

## 2 Related Work

### 2.1 Federated Learning

**Mislabeling.** In the context of distributed learning, there may be instances where nodes misclassify certain data, leading to a decrease in data quality. Some research further extends this issue to Byzantine attacks in distributed learning Shi et al. (2021); Fang et al. (2020); Shejwalkar & Houmansadr (2021); Cao et al. (2020). In these attacks, malicious nodes can manipulate their model parameters (such as weights or gradients) to degrade the accuracy of the global model. Various strategies have been proposed to defend against Byzantine attacks in distributed learning So et al. (2020). These include client selection strategies, score-based detection methods, spectral-based outlier detectors, and update denoising. In our DISTDD, we also consider that each client's data may have bad quality since the clients' labels might be wrong.

**Hyper-parameter optimization.** Previous researchers also have worked on hyper-parameter optimization in FL. Zhou et al. (2021) leverages meta-learning techniques to utilize local and asynchronous to optimize the hyper-parameter. Khodak et al. (2020) applied techniques from NAS with weight-sharing to FL with personalization to modify local training-based FL. Agrawal et al. (2021) clusters edge devices based on the training hyper-parameters and genetically modifies the parameters cluster-wise. However, these approaches still require multiple communication processes.

**Key contributions for federated learning.** The debugging of a typical FL process requires iteratively choosing different parameters, thus bringing heavy communication costs. However, our proposed method, DISTDD, distillates the data from distributed clients rather than just directly training a single classifier model. By acquiring this distilled dataset, FL servers can iteratively train the global model on the distilled dataset locally without iteratively debugging the parameters of FL, thus avoiding high communication costs.

### 2.2 Dataset Distillation

Dataset distillation Wang et al. (2020) involves creating a condensed dataset from a larger one, with the aim of training models to achieve strong performance on the original extensive dataset. This distillation algorithm takes a substantial real-world dataset as input (the training set) and generates a compact, synthetic distilled dataset. The production of high-quality, compact, distilled datasets is significant for enhancing dataset comprehension and a wide array of applications, including continual learning, safeguarding privacy, and optimizing neural architecture in tasks such as neural architecture search.

Some previous works aim to use gradient or trajectory-matching surrogate objectives to achieve distillation. Shin et al. (2023); Du et al. (2023); Cazenavette et al. (2022) use trajectory matching to distill dataset. Zhao & Bilen (2021); Liu et al. (2023b) propose using gradient matching to distill the dataset. Wang et al. (2022); Zhao & Bilen (2023); Zhao et al. (2023) align the features condense dataset, which explicitly attempts to preserve the real-feature distribution as well as the discriminant power of the resulting synthetic set, lending itself to strong generalization capability to various architectures. Bohdal et al. (2020); Sucholutsky & Schonlau (2021) propose simultaneously distilling both images and their labels, thus assigning each synthetic sample a 'soft' label rather than a 'hard' label.

Data distillation has been employed as an effective strategy to enhance the performance of distributed learning systems or to develop novel distributed learning architectures Zhang et al. (2022); Pi et al. (2023); Xiong et al. (2023); Song et al. (2023). These studies have demonstrated how data distillation can be

integrated into distributed learning to achieve more effective learning outcomes or to introduce innovative architectural paradigms within the distributed learning framework.

**Key contributions for dataset distillation.** DISTDD introduces a new scene for dataset distillation. In this scene, a central server wants to distill the data from distributed clients. This scene brings new challenges in protecting each distributed client's privacy and cutting communication costs.

## 3 Methodology

---

**Algorithm 1** Federated Data Distillation through gradient matching

---

1: **Input**: A central server $p$, distributed clients $i = 0, ..., I - 1$. The portion $\delta$ of selected participated clients per round. Training set $\mathcal{T}_i$ for each client $i = 0, ..., I - 1$. Randomly set of synthetic samples $S$ for $C$ classes, probability distribution over randomly weights $P_{\theta_0}$, neural network $\phi_\theta$, number of loop steps $T$, number of steps for updating weights $\varsigma_\theta$ and synthetic samples $\varsigma_S$ in each inner-loop step respectively, learning rates for updating weights $\eta_\theta$ and synthetic samples $\eta_S$.
2: Initialize $\theta_0 \sim P_{\theta_0}$          ▷ Neural networks initialization.
3: **for all** $t = 0, ..., T - 1$ **do**
4:      $p$ sends $\theta_t$ to $I$
5:      $p$ samples $\delta \times I$ clients $I'$ from $I$          ▷ Participant selection.
6:      **for all** $c = 0, ..., C - 1$ **do**
7:          **for all** $i = 0, ..., I' - 1$ **do**
8:              Each client $i$:
9:              $i$ samples a mini-batch $B_c^{\mathcal{T}_i} \sim \mathcal{T}_i$
10:              $\mathcal{L}_c^{\mathcal{T}_i} = \frac{1}{\left|B_c^{\mathcal{T}_i}\right|} \sum_{(\boldsymbol{x}, y) \in B_c^{\mathcal{T}_i}} \ell\left(\phi_{\boldsymbol{\theta}_t}(\boldsymbol{x}), y\right)$          ▷ Update local neural networks.
11:              $i$ computes $g_{t,c,i} = \nabla_{\boldsymbol{\theta}} \mathcal{L}_c^{\mathcal{T}_i}(\boldsymbol{\theta}_t)$          ▷ Compute updated gradients.
12:              $i$ sends $g_{t,c,i}$ to $p$
13:          **end for**
14:          $p$ computes $G_{t,c} = \sum_{i=0}^{I'-1} g_{t,c,i}$          ▷ Aggregate global gradients.
15:          $p$ samples a mini-batch $B_c^S \sim S$
16:          $p$ computes $\mathcal{L}_c^S = \frac{1}{|B_c^S|} \sum_{(\boldsymbol{s}, y) \in B_c^S} \ell\left(\phi_{\boldsymbol{\theta}_t}(\boldsymbol{s}), y\right)$
17:          $p$ computes $\mathfrak{g}_{t,c} = \nabla_\theta \mathcal{L}_c^S$
18:          $\mathcal{S}_c \leftarrow opt-\text{alg}_\mathcal{S}\left(D\left(G_{t,c}, \mathfrak{g}_{t,c}\right), \varsigma_\mathcal{S}, \eta_\mathcal{S}\right)$          ▷ Update distilled dataset.
19:      **end for**
20:      $p$ updates $\theta_{t+1} \leftarrow G_{t,c}$          ▷ Update neural networks.
21: **end for**
22: **Output**: $S$

---

In DISTDD, there is a central server $p$. And there are multiple distributed clients $i = 0, ..., I - 1$, each has a local dataset $\mathcal{T}_i$. The dataset contains $C$ classes.

To get the optimal parameter for training, $p$ has to optimize the hyper-parameter of FL by repeating the whole FL process. However, due to FL's high communication and computation costs, it is inefficient for $p$ and $I$ to conduct such a costly process. Thus, it is more reliable for $p$ to distill the datasets from the client set $I$ into one distilled dataset and use the distilled dataset to optimize the parameters. However, it is unfeasible for $p$ to collect the datasets from all the clients and do the data distillation locally on the server. Thus, DISTDD achieves the data distillation in a distributed way:

Initially, $p$ randomly generate an initialized set of synthetic samples $S$ containing $C$ classes, probability distribution over randomly initialized weights $P_{\theta_0}$. $p$ also initialize a deep neural network $\phi_\theta$, which serves as a classifier for this dataset. Now $p$ set the number of loop steps $T$, the number of steps for updating weights $\varsigma_\theta$ and synthetic samples $\varsigma_S$ in each inner-loop step respectively, learning rates for updating weights $\eta_\theta$ and synthetic samples $\eta_S$.

In each iteration, $p$ will first sends the classifier model weight $\theta_t$ to each client $i$. Each client $i$ samples a mini-batch $B_c^{\mathcal{T}_i} \sim \mathcal{T}_i$ from its local dataset $\mathcal{T}_i$. And the mini-batch $B_c^{\mathcal{T}_i}$ will be used to compute the loss $\mathcal{L}_c^{\mathcal{T}_i}$ using the classifier $\theta_t$:

$$\mathcal{L}_c^{\mathcal{T}_i} = \frac{1}{\left| B_c^{\mathcal{T}_i} \right|} \sum_{(\boldsymbol{x}, y) \in B_c^{\mathcal{T}_i}} \ell \left( \phi_{\boldsymbol{\theta}_t}(\boldsymbol{x}), y \right). \tag{1}$$

$i$ then computes the gradient $g_{t,c,i}$ using the loss as:

$$g_{t,c,i} = \nabla_{\boldsymbol{\theta}} \mathcal{L}_c^{\mathcal{T}_i} (\boldsymbol{\theta}_t) \tag{2}$$

and sends the gradient $g_{t,c,i}$ back to $p$, extracting each client's data knowledge into the gradient $g_{t,c,i}$.

After receiving the gradient $g_{t,c,i}$ from sampled clients, $p$ aggregate all the gradients to a global gradient as $G_{t,c}$:

$$G_{t,c} = \sum_{i=0}^{I-1} g_{t,c,i} = \sum_{i=0}^{I-1} \nabla_{\boldsymbol{\theta}} \mathcal{L}_c^{\mathcal{T}_i} (\boldsymbol{\theta}_t). \tag{3}$$

The central server aggregates all the clients' knowledge about their local data into the global gradient $G_{t,c}$ by aggregating all the gradients $g_{t,c,i}$ from each client $i$. To be noted, each client only sends the gradient update $g_{t,c,i}$ to the central server without sending other privacy-related information, thus achieving the same level of privacy as the typical FL process.

Then $p$ samples a mini-batch $B_c^S \sim S$ from the synthetic dataset $S$, and computes the gradient using the classifier $\theta_t$ as

$$\mathfrak{g}_{t,c} = \nabla_{\theta} \mathcal{L}_c^{\mathcal{S}} = \nabla_{\theta} \mathcal{L}_c^{\mathcal{S}} = \frac{1}{|B_c^{\mathcal{S}}|} \sum_{(\boldsymbol{s}, y) \in B_c^{\mathcal{S}}} \ell \left( \phi_{\boldsymbol{\theta}_t}(\boldsymbol{s}), y \right). \tag{4}$$

$p$ will compute the loss $D(G_{t,c}, \mathfrak{g}_{t,c})$ by computing the gradient mismatching between $G_{t,c}$ and $\mathfrak{g}_{t,c}$ as:

$$D(G_{t,c}, \mathfrak{g}_{t,c}) = D \left( \nabla_{\boldsymbol{\theta}} \mathcal{L}_c^{\mathcal{S}} (\boldsymbol{\theta}_t), \nabla_{\boldsymbol{\theta}} \mathcal{L}_c^{\mathcal{T}} (\boldsymbol{\theta}_t) \right) \tag{5}$$

Then $p$ update the synthetic data $\mathcal{S}_c$ of class $c$ by matching the loss as

$$\begin{aligned} \mathcal{S}_c &\leftarrow opt - \mathrm{alg}_{\mathcal{S}} \left( D(G_{t,c}, \mathfrak{g}_{t,c}), \varsigma_{\mathcal{S}}, \eta_{\mathcal{S}} \right) \\ &= opt - \mathrm{alg}_{\mathcal{S}} \left( D \left( \nabla_{\boldsymbol{\theta}} \mathcal{L}_c^{\mathcal{S}} (\boldsymbol{\theta}_t), \nabla_{\boldsymbol{\theta}} \mathcal{L}_c^{\mathcal{T}} (\boldsymbol{\theta}_t) \right), \varsigma_{\mathcal{S}}, \eta_{\mathcal{S}} \right) \end{aligned} \tag{6}$$

This step aims to update the synthetic data $\mathcal{S}_c$ by computing gradient mismatch.

At the last of each iteration, $p$ updates the model weight $\boldsymbol{\theta}_{t+1}$ as

$$\boldsymbol{\theta}_{t+1} \leftarrow opt - a \lg_{\boldsymbol{\theta}} \left( \mathcal{L}^{\mathcal{S}} (\boldsymbol{\theta}_t), \varsigma_{\boldsymbol{\theta}}, \eta_{\boldsymbol{\theta}} \right) \tag{7}$$

After $T$ iterations, the datasets distributing across the set of the clients $I$ are distilled into a dataset labeled as $S$.

### 3.1 Protect Privacy

However, there are still many claims about the privacy of federated learning. Previous researchers claim that the exchanged gradient updates between clients and the central server can still leak privacy-related information from clients to the central server. Furthermore, we consider providing more privacy protection methods for DISTDD by introducing DPSGD Abadi et al. (2016) into our DISTDD framework.

The DPSGD is performed as: For each $x_j$ in mini-batch $B_c^{\mathcal{T}_i}$, the gradient is computed as

$$g_{t,c,i}(x_j) = \nabla_{\theta} \mathcal{L}_c^{\mathcal{T}_i} (\theta_t, x_j). \tag{8}$$

Then DISTDD applies clip gradient as

$$\bar{g}_{t,c,i}(x_j) \leftarrow g_{t,c,i}(x_j) / \max \left( 1, \frac{\|g_{t,c,i}(x_j)\|_2}{C} \right). \tag{9}$$

| Datasets | MNIST | | | FashionMNIST | | | CIFAR | | |
|---|---|---|---|---|---|---|---|---|---|
| Dir | 1 | 0.5 | 0.1 | 1 | 0.5 | 0.1 | 1 | 0.5 | 0.1 |
| Whole Dataset | | 99.3 | | | 93.6 | | | 87.2 | |
| GM (IPC=100) | | 97.2 | | | 91.1 | | | 80.7 | |
| FedAvg | 98.8 | 94.3 | 92.5 | 92.5 | 86.3 | 74.3 | 86.3 | 75.3 | 65.9 |
| FedProx | 99.1 | 95.7 | 93.8 | 92.6 | 90.7 | 86.2 | 86.5 | 80.1 | 72.1 |
| DistDD (IPC=10) | 94.1 | 90.3 | 82.3 | 82.1 | 75.3 | 67.6 | 51.3 | 47.4 | 41.5 |
| DistDD (IPC=50) | 95.6 | 92.8 | 83.1 | 84.7 | 82.5 | 69.3 | 75.2 | 62.4 | 51.5 |
| DistDD (IPC=100) | 96.9 | 93.2 | 84.0 | 90.1 | 84.3 | 72.5 | 78.2 | 69.2 | 57.3 |

Table 1: The performance comparison to the whole dataset (centralized training using the whole dataset), GM (centralized gradient matching using the whole dataset), FedAvg, FedProx, and DistDD. For FedAvg and DistDD, we set the client number to 50. The FedAvg's accuracy value is compared with the whole dataset training and recorded in the table. The DISTDD's accuracy is compared with the gradient matching's accuracy (IPC=100) and recorded in the table. The comparison between (FedAvg-Whole dataset) and (DISTDD - GM) indicates that DISTDD's performance aligns with FedAvg on the distributed pattern.

Then we add differential privacy noise to it as

$$\tilde{g}_{t,c,i} \leftarrow \frac{1}{|B_c^{\mathcal{T}_i}|} \left( \sum_j \overline{g}_{t,c,i}(x_j) + \mathcal{N}\left(0, \sigma^2 C^2 \mathbf{I}\right) \right) \tag{10}$$

### 3.2 Tackle with Mislabeling Problem

The issue of mislabeling in data distillation, particularly in distributed learning systems, poses a significant challenge to the integrity and effectiveness of machine learning models. This problem arises when clients contributing to the distilled dataset inadvertently or intentionally introduce errors in labeling, thereby compromising the data quality. Such inaccuracies can significantly impact the performance of the aggregated dataset, especially in scenarios where diverse clients contribute data, increasing the likelihood of inconsistencies and errors. Addressing this issue requires a focus on developing robust methods for detecting the potential mislabeling of clients and discarding their gradient updates.

We introduce the Median from Yin et al. (2018) to tackle with mislabeling problem in DISTDD:

**Definition (Coordinate-wise median).** For vectors $x^i \in P, i \in [I]$, the coordinate-wise median $g :=$ $med\{x^i : i \in [I]\}$ is a vector with its $k$-th coordinate being $g_k = med\{x_k^i : i \in [I]\}$ for each $k \in [d]$, where med is the usual (one-dimensional) median.

## 4 Evaluation

We first reveal our experiment setting in §4.1. Next, we compare DISTDD with FedAvg schemes in §4.2. Then, we consider the mislabeling situations and evaluate the DISTDD method under different portions of mislabeling clients in §4.3. The data distribution problem of nonIID is considered in §4.4. To prove the effectiveness of using DISTDD in the use cases, we evaluated DISTDD under NAS settings in §4.5.

### 4.1 Experiment Setting

We discussed our experiment settings in this section:

- **Models.** In our experimental setup, we employ a Convolutional Neural Network (ConvNet) architecture as the foundational network for our study.
- **Datasets.** We leverage three image classification datasets, namely MNIST, FashionMNIST, and CIFAR-10, as the experimental datasets.

- **Client number.** The default configuration for our system includes a predefined client count of 20. Furthermore, our system employs a randomized participant selection process, wherein 50% of the clients actively participate in the training process during each iteration (this setting follows the convention of both FL and DD).
- **Image per class.** Notably, we adhere to a predefined standard of 100 images per class as the default quantity for each image category.
- **Communication round.** Our experimentation proceeds throughout 500 communication rounds, with each round representing a critical iteration in the distributed learning process. To measure the efficacy of our system, we employ the classification accuracy of the base model trained on the generated images as the principal metric for evaluation.
- **Data distribution.** We used the Dirichlet distribution to guide the data segmentation process in this experiment. The Dirichlet distribution is a multinomial distribution often used to represent the probability distribution of multiple categories and is very suitable for simulating the distribution of different categories in a data set. The concentration parameter $\alpha$ of the Dirichlet distribution (we noted as *dir*) is used to guide the non-iid degree of the distribution.

## 4.2 Comparison with FedAvg

**Methods.** The comparative analysis presented in Table 1 evaluates the performance of DISTDD with other schemes, including the whole dataset (centralized training using the whole dataset), GM (centralized gradient matching using the whole dataset), FedAvg, FedProx under various IPC settings and data distribution scenarios among distributed clients. This analysis specifically focuses on the impact of the image per class (IPC) parameter on DISTDD and how it compares to the performance of other schemes.

**Results.** The results show that DISTDD performs similarly to FedAvg as the IPC value increases, with a notable performance gap when IPC is below 100. At IPC=100, DISTDD matches FedAvg's performance. In IID settings with Dirichlet distribution parameter (dir) of 1, both systems exhibit identical accuracy. However, in highly non-IID scenarios (dir=0.1), DISTDD underperforms significantly compared to FedAvg, highlighting its weaker robustness in diverse data environments.

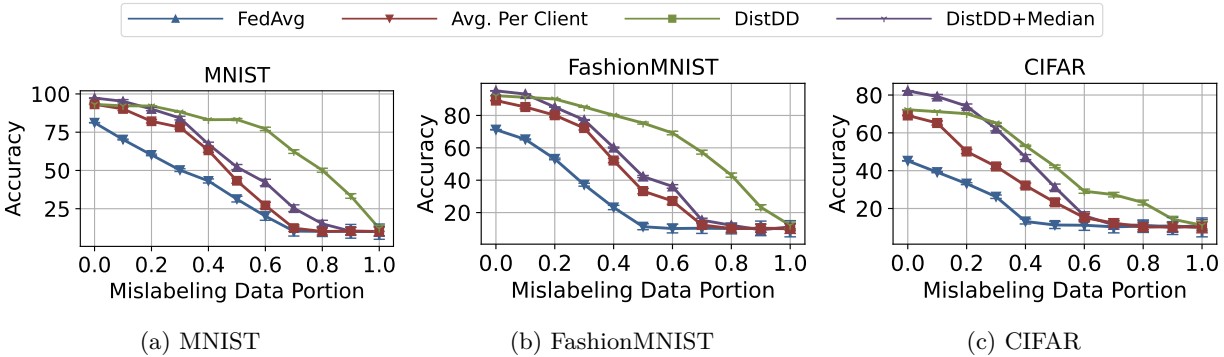

(a) MNIST  (b) FashionMNIST  (c) CIFAR

Figure 2: We conducted further study of mislabeling in data. Considering developers may mistakenly or maliciously mislabel the input dataset, we evaluated DISTDD's performance under this situation. The mislabeling data portion is set from 0.0 to 1.0.

## 4.3 Mislabeling Situation

**Methods.** We then investigate the impact of mislabeling in data distillation, where clients may introduce labeling errors that compromise the integrity of the distilled data. We model this as clients being prone to a certain proportion of mislabeled samples, with consistent mislabeling patterns across clients. We compare the performance of four frameworks: FedAvg, local gradient matching, DISTDD and a DISTDD adaptation of the Median anti-byzantine attack defense (Yin et al. (2018)) to address mislabeling challenges.

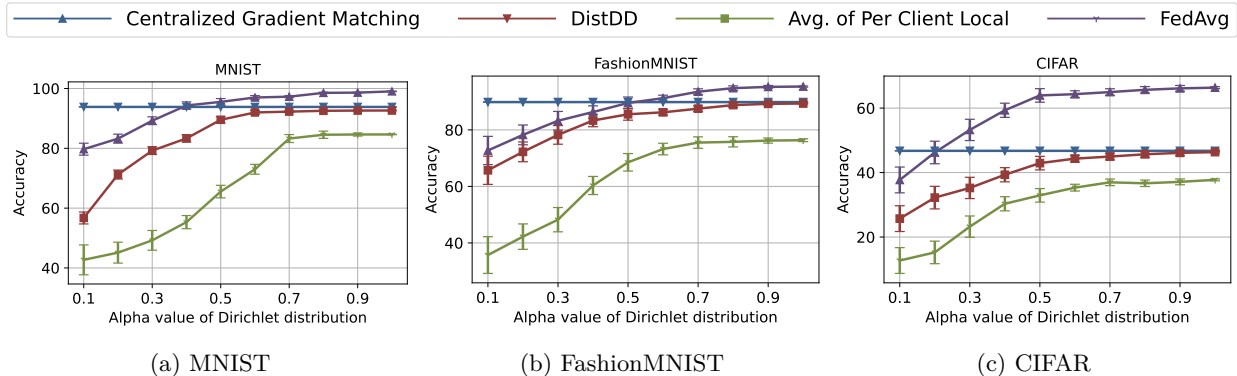

(a) MNIST       (b) FashionMNIST       (c) CIFAR

Figure 3: Considering the non-iid nature of federated learning, we studied how the non-iid distribution affects the performance of DISTDD. We use the Dirichlet distribution to model the non-iid distribution.

**Results.** The outcomes of the evaluation are presented in Figure 2. Our findings reveal that local gradient matching in its raw form is ill-equipped to counter this threat, thereby leading to a degradation in performance as the proportion of mislabeled data increases. DISTDD on the other hand, demonstrates a certain degree of resilience in specific scenarios due to its ability to aggregate knowledge from diverse clients, thus mitigating the effects of the mislabeling issue to some extent. Notably, when augmented with the Median mechanism, DISTDD exhibits a robust defense against mislabeling, even in the presence of widespread mislabeling, resulting in consistently high levels of accuracy. Also, we compare FedAvg with our DISTDD. The results show that DISTDD with Median as the defense method can overcome the FedAvg scheme without any defense.

## 4.4 Non-iid Situation

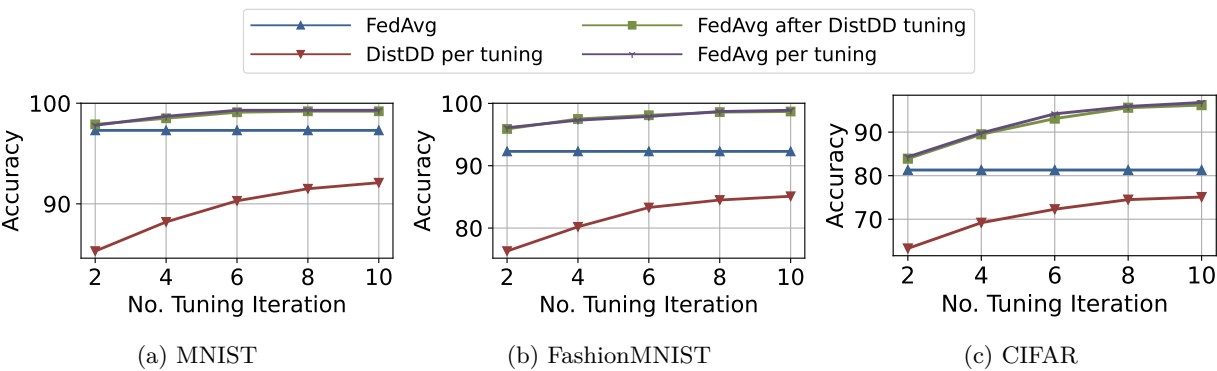

(a) MNIST       (b) FashionMNIST       (c) CIFAR

Figure 4: Use case evaluation for DISTDD. The results indicate that using DISTDD for Network Architecture Search (NAS) over Federated Learning (FL) is as effective as the traditional FedAvg approach in terms of accuracy. However, DISTDD offers a significant advantage in reducing time costs, especially as the number of tuning iterations increases. This is because, unlike FedAvg, DISTDD requires less communication after the initial tuning, presenting a more efficient trade-off between time and performance.

**Methods.** This experiment examines the impact of non-iid data distribution on DISTDD's classification accuracy, which is crucial for assessing its performance in real-world scenarios with diverse client data patterns. Using the Dirichlet distribution to simulate non-iid conditions, we vary the parameter alpha (dir) from 0.1 to 1.0 to control the degree of non-iid. To evaluate their performance under these conditions, a comparative analysis is conducted between centralized gradient matching, DistDD, local gradient matching, and FedAvg.

**Results.** The results shown in Figure 3, indicate that DISTDD performs significantly worse than centralized gradient matching in highly non-iid scenarios. However, as the data distribution becomes nearly iid, DISTDD approaches the performance of centralized gradient matching. Additionally, we evaluate per-client local gradient matching, which shows reduced efficacy in non-iid settings, as demonstrated by the accuracy results in Figure 3.

## 4.5 Use Case for DistDD

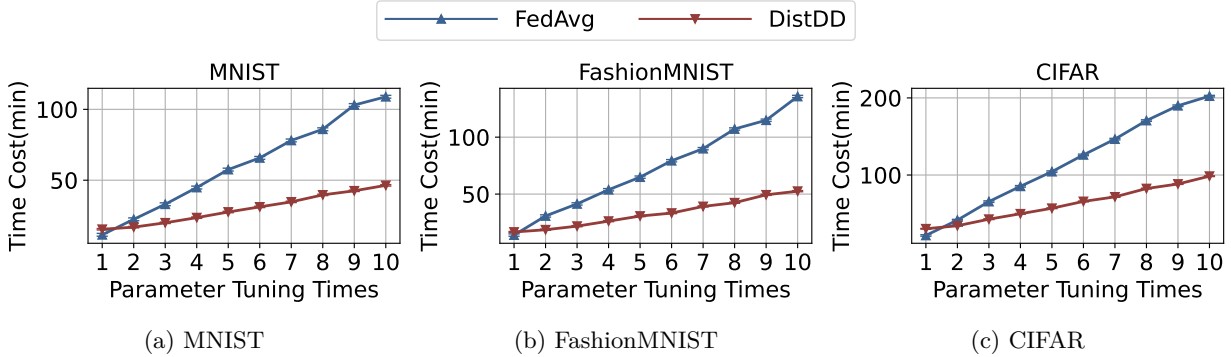

(a) MNIST       (b) FashionMNIST       (c) CIFAR

Figure 5: Time overhead comparison between FedAvg and DISTDD under different hyper-parameter tuning times.

**Methods.** To prove DISTDD's effectiveness on the use case B: NAS over FL, we provided an example evaluation as shown in Figure 4. We compared original FedAvg accuracy, DISTDD's accuracy in each tuning iteration (after the DistDD's distilled dataset-based NAS, the network was trained on DISTDD's distilled dataset.), FedAvg's accuracy after DistDD tuning (after DistDD's distilled dataset-based NAS, the network was trained again using FedAvg) and FedAvg tuning (directly using FedAvg for NAS). We also compare DISTDD's time cost with FedAvg's time cost under increasing parameter tuning times (see Figure 5).

**Results.** The results show that FedAvg after DISTDD NAS has a similar accuracy with FedAvg for NAS. This proves DistDD's effectiveness for the NAS over FL. When only searching for the architecture for one time, the two frameworks' time costs are nearly the same. While, as the tuning periods increase, FedAvg's time cost goes above DISTDD's time cost soon. This is because DISTDD does not need to communicate for the tuning process after the 1st tuning process.

## 5 Ablation Study

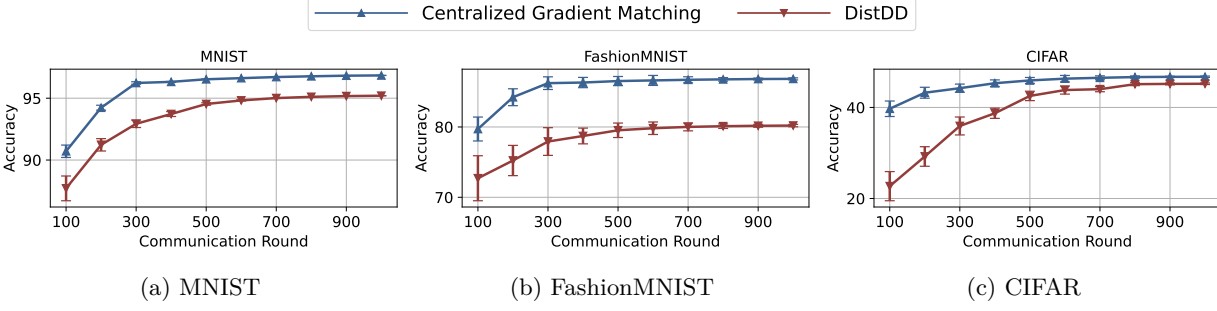

(a) MNIST       (b) FashionMNIST       (c) CIFAR

Figure 6: Ablation study of different Communication Rounds.

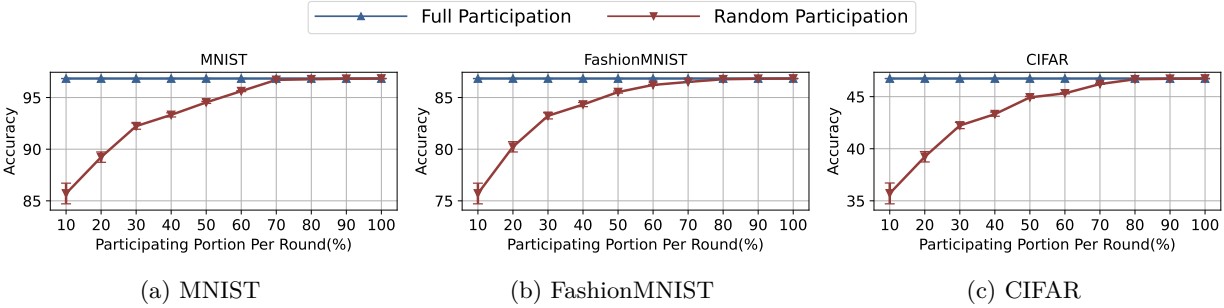

Figure 7: Ablation study of different participants portion.

## 5.1 Different Nodes Number

We evaluate the performance of DISTDD in response to varying degrees of node participation. In this particular experiment, it is notable that the cumulative volume of data samples across all clients remains unaltered. Consequently, as we increase the number of participating nodes, the number of data samples allocated to each individual client simultaneously diminishes. We rely on the classification accuracy outcomes to illuminate the performance changes, as shown in Figure 12.

To conduct the comparative analysis, we compare three distinct configurations: firstly, the local gradient matching; secondly, DISTDD featuring full participation from all nodes; and thirdly, DISTDD with a 50% random client participation scheme. The experiment results manifest a notable trend. Specifically, the performance of DISTDD with full participation exhibits a gradual decline with the amplification of node numbers; nonetheless, this decline is relatively modest. In contrast, the performance of DISTDD with random participation shows a substantially steeper descent in accuracy.

## 5.2 Image number per class

In this section, we explore the impact of the number of generated images per class with a specific focus on its effect on classification accuracy. To undertake this ablation study, we systematically vary the quantity of images per class, encompassing the values 1, 10, 20, 30, 40, and 50. The outcomes are shown in Figure 13.

It is notable that local gradient matching reaches convergence primarily when the image count per class ranges between 10 and 20. In contrast, DISTDD exhibits a convergence behavior at a significantly higher threshold, typically exceeding 30 images per class. This observation suggests that DISTDD necessitates a more substantial quantity of images to aggregate knowledge from the distributed clients effectively. However, it is noteworthy that the performance of DISTDD demonstrates the potential to approximate the performance levels achieved by local gradient matching when the image count per class reaches sufficiently high values. This disparity in the requisite image count for DISTDD may be attributed to the expansive dispersion of data across numerous clients, consequently mandating a greater number of generated images to facilitate convergence.

## 5.3 Communication Rounds

In this section, we evaluate the influence of communication rounds on the performance of DISTDD with a particular emphasis on its impact on classification accuracy. We conduct this analysis by contrasting two configurations of DISTDD one with full client participation per round and another with random participation of 50% of the clients per round, within the context of a 20-client scenario. The results, illustrated in Figure 10, offer the observed effects.

Evidently, DISTDD with full client participation typically requires approximately 300 communication rounds to converge. In contrast, the variant of DISTDD featuring random client participation necessitates a sig-

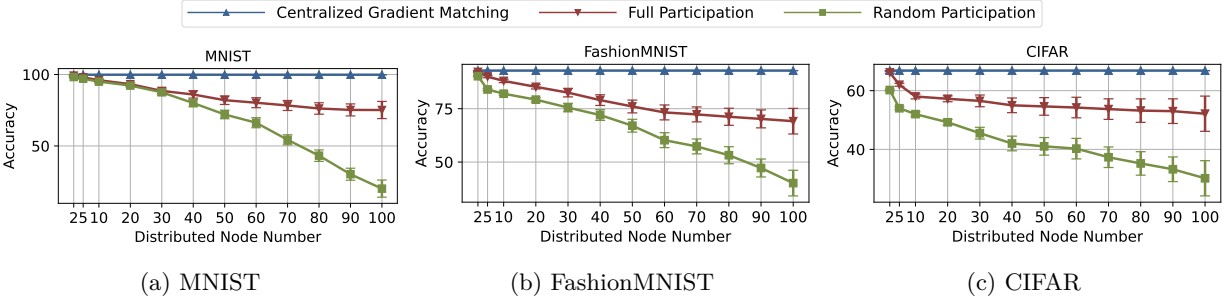

Figure 8: Ablation study of node number.

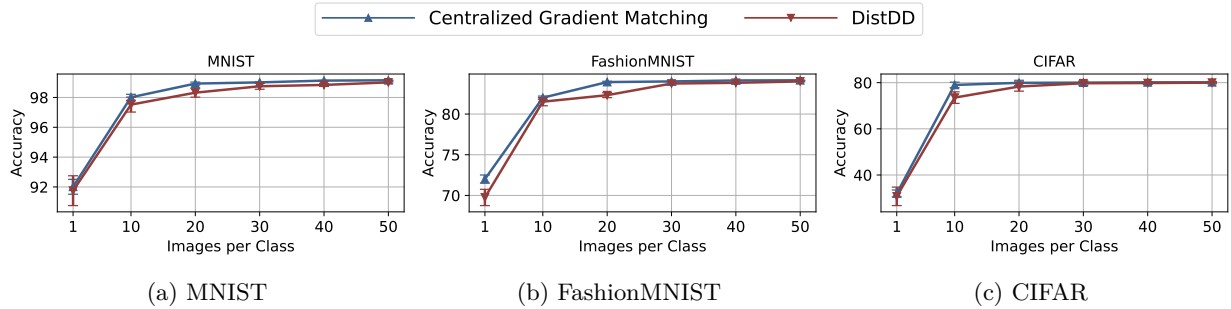

Figure 9: Ablation study of different image numbers per class.

nificantly greater number of communication rounds to achieve the same convergence. This discrepancy in the convergence rate primarily stems from random client participation, which mandates a more extended communication process for each client to convey and synchronize their knowledge with the central server.

## 5.4 Portion of Selected Clients per Round

Next, we study the effect of the proportion of selected clients per round, focusing on random participation throughout 500 communication rounds. We maintain a constant client count of 20 while adhering to a Dirichlet distribution parameter ($dir = 1.0$) for data partitioning. The proportion of participating clients is systematically varied, ranging from 10% to 100% (representing full participation).

Noteworthy is the observation that it necessitates a participation rate of 80% within the random selective participation scheme to achieve parity in classification accuracy with full participation. Conversely, when the participation rate falls below the 50% threshold, the performance of DISTDD markedly falls behind that of local gradient matching. This disparity in performance underlines the significance of the participation proportion in the context of random selection and underscores the trade-off between participation rate and classification accuracy.

## 6 Discussion

**The same level of privacy protection:** FL has been widely considered an efficient method to aggregate knowledge from distributed clients and protect distributed clients' privacy. Although FL has many privacy challenges, the privacy level itself is enough for many scenes. Like FL, our proposed method DISTDD only allows the gradient updates exchange between clients and servers. This gradient update is used in the central server's gradient matching process to construct a distilled dataset. There is no other privacy-related

information exchanged in DISTDD. Thus, DISTDD, as an alternative to FL, can protect privacy to the same level as FL.

**Abstract for global dataset**. In fact, DISTDD provides the abstract for the global dataset. By performing the gradient matching in a distributed way, DISTDD aggregates the global knowledge into the distilled dataset as a global abstract. This abstract enables the server of FL to tune the parameter and the architecture without high communication costs.

# 7 Conclusion

In conclusion, our work introduces a new distributed data distillation framework, named DISTDD (Distributed Data Distillation through gradient matching), which combines the gradient matching methods with distributed learning. This novel approach enables the extraction of distilled knowledge from a diverse set of distributed clients, offering a solution for aggregating large-scale distributed data while enabling the server to train the global model on the global dataset freely without concern about communication overhead. Our comprehensive experimentation has also demonstrated the robustness and effectiveness of DISTDD in various scenarios.

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

# A   Convergence Analysis

We formulate the proof of our DISTDD as two steps: First, we prove the convergence of our FL process. Then, we prove the convergence of the gradient matching process by proving that the synthetic dataset can be very close to the original dataset.

## A.1   Problem Formulation

We formulate local SGD as follows:

$$\theta_t^{i,k+1} := \theta_t^{i,k} - \eta \frac{\partial \mathcal{L}}{\partial \theta_t^{i,k}} = \theta_t^{i,k} - \eta \bigtriangledown \theta_t^{i,k} \tag{11}$$

$\theta^i$ is the local model parameter for client $i$, $t$ is global round index and $k$ is local step index.

And we consider the overall optimization objective as

$$minF(\theta) = \mathbb{E}_{i \sim \mathcal{C}}(F_i(\theta)) \tag{12}$$

We have a client population as $\mathcal{C} = 1, 2, 3, ..., M$.

## A.2   Assumptions

To prove the convergence of our work, we have two main assumptions.

**Assumption 1: Unbiased stocahstic gradient**. The expectation of the stochastic gradient for a given $\theta_i^{t,k}$ is equal to the average local gradient for a given model $\phi(\cdot)$. This is to say, the gradient expectation of the SGD equals the gradient of the GD:

$$\mathbb{E}[\bigtriangledown \theta_t^{i,k} | \theta_t^{i,k}] = \bigtriangledown F_i(\theta_t^{i,k}) \tag{13}$$

Given a dataset of $\mathcal{T}_i = (\S i, \dagger_i) i = 0^N$, where the N denotes the length of the whole dataset. The objective of the GD and its gradients are calculated as:

$$F_i(x_t^{(i,k)}) = \frac{1}{N} \sum_{i=1}^{N} \mathcal{L}(\phi(\S_i), \dagger_i)$$

$$\nabla F_i(x_t^{(i,k)}) = \frac{1}{N} \sum_{i=1}^{N} \nabla \mathcal{L}(\phi(\S_i), \dagger_i) \tag{14}$$

In this case, the expectation is the weighted average of a single batch with batch size as *bn*, i.e.,

$$\mathbb{E}\left[\nabla x_t^{(i,k)} | x_t^{(i,k)}\right]$$
$$= \sum_{j=1}^{N-bn+1} \left( \sum_{i=1}^{bn} \frac{\partial \mathcal{L}}{\partial x_{t,s_j}^{(i,k)}} \cdot P(I = i | S = s_j) \right) \cdot P(S = s_j)$$
$$= \sum_{j=1}^{N-bn+1} \left( P(I = i | S = s_j) P(S = s_j) \sum_{i=1}^{bn} \frac{\partial \mathcal{L}}{\partial x_{t,s_j}^{(i,k)}} \right)$$
$$= \frac{1}{N} \sum_{i=1}^{N} \frac{\partial \mathcal{L}}{\partial x_t^{(i,k)}}$$
$$= \frac{1}{N} \sum_{i=1}^{N} \nabla \mathcal{L}(\phi(\S_i), \dagger_i)$$
$$= \nabla F_i(x_t^{(i,k)}) \tag{15}$$

where batch set is $S = s_1, \cdots, s_{bn}$. SGD or Adam is a stochastic optimization algorithm that randomly selects samples from the batch for gradient calculation.

**Assumption 2: Bounded variance:**

$$\mathbb{E}\left[\left\|\nabla x_t^{(i,k)} - \nabla F_i(x_t^{(i,k)})\right\|^2 | x_t^{(i,k)}\right] \leq \sigma^2 \tag{16}$$

This is to say the gradient of the SGD is close to that of the GD.

**Assumption 3: L-Smooth:** Local gradient $\nabla F_i(x)$ and global gradient $\nabla F(x)$ is $\zeta$-uniformly bounded.

$$\max_l \sup_x \left\|\nabla F_i(x_t^{(i,k)}) - \nabla F(x_t^{(i,k)})\right\| \leq \zeta \tag{17}$$

### A.3 Proof

Generally, we want to prove that

$$\|F(x_t^{k+1}) - F(x^*)\| \\ \leq \|F(x_t^k) - F(x^*)\|, \forall t, k \in [1, 2, 3, \cdots] \tag{18}$$

where $F(x^*)$ is the optimal. Or, we give a weaker claim:

$$\mathbb{E}\left[\frac{1}{\tau T}\sum_{t=0}^{T-1}\sum_{k=1}^{\tau} F\left(\overline{\boldsymbol{x}}_t^k\right) - F\left(\boldsymbol{x}^\star\right)\right] \tag{19}$$
$$\leq \text{ an upper bound decreasing with } T.$$

Note that $\mathbb{E}$ in this paper denotes $\mathbb{E}_{i\sim\mathcal{C}}$, where $\mathcal{C}$ denotes the client set. Therefore, we can say that the $\mathbb{E}$ is generally calculating the expectation over all the clients.

**Decentralized optimization**: Originating from the decentralized optimization, we derive the shadow sequence to indicate the update process.

$$\overline{x}_t^k := \frac{1}{M}\sum_{i=1}^{M} x_t^{(i,k)} \tag{20}$$

Then, at round $t$ local epoch $k + 1$,

$$\overline{x}_t^{k+1} = \overline{x}_t^k - \eta\frac{1}{M}\sum_{i=1}^{M} x_t^{(i,k)} \tag{21}$$

Then, we want to prove two lemmas:

**Lemma 1: (Per Round Progress)** Assuming the client learning rate satisfies $\eta < \frac{1}{4L}$, we prove that the expectation for each round is bounded.

$$\mathbb{E}\left[\frac{1}{\tau}\sum_{k=1}^{\tau} F\left(\overline{\boldsymbol{x}}_t^k\right) - F\left(\boldsymbol{x}^\star\right)\bigg|\mathcal{F}^{(t,0)}\right]$$
$$\leq \frac{1}{2\eta\tau}\left(\left\|\overline{\boldsymbol{x}}_t^0 - \boldsymbol{x}^\star\right\|^2 - \mathbb{E}\left[\left\|\overline{\boldsymbol{x}}_t^\tau - \boldsymbol{x}^\star\right\|^2 \mid \mathcal{F}^{(t,0)}\right]\right) \tag{22}$$
$$+ \frac{\eta\sigma^2}{M} + \frac{L}{M\tau}\sum_{i=1}^{M}\sum_{k=0}^{\tau-1}\mathbb{E}\left[\left\|\boldsymbol{x}_t^{(i,k)} - \overline{\boldsymbol{x}}_t^k\right\|^2 \mid \mathcal{F}^{(t,0)}\right]$$

where $\mathcal{F}^{(t,0)}$ is the $\sigma$-field representing all the historical information up to the start of the $t$-th round

**Lemma 2 (Bounded Client Drift):** Assuming the client learning rate satisfies $\eta < \frac{1}{4L}$, we prove that the Bound in the lemma 1 is decreasing with $T$.

$$\mathbb{E}\left[\left\|\boldsymbol{x}_t^{(i,k)} - \overline{\boldsymbol{x}}_t^k\right\|^2 \mid \mathcal{F}^{(t,0)}\right] \leq 18\tau^2\eta^2\zeta^2 + 4\tau\eta^2\sigma^2 \tag{23}$$

where $\mathcal{F}^{(t,0)}$ is the $\sigma$-field representing all the historical information up to the start of the $t$-th round

**Theorem 1 (Convergence Rate for Convex Local Functions):** Under the aforementioned assumptions $(a) - (g)$, if the client learning rate satisfies $\eta < \frac{1}{4L}$, then one has

$$
\begin{aligned}
&\mathbb{E}\left[\frac{1}{\tau T}\sum_{t=0}^{T-1}\sum_{k=1}^{\tau} F\left(\overline{\boldsymbol{x}}_t^k\right) - F\left(\boldsymbol{x}^\star\right)\right]\\
&\leq \frac{\mathbb{D}^2}{2\eta\tau T} + \frac{\eta\sigma^2}{M} + 4\tau\eta^2 L\sigma^2 + 18\tau^2\eta^2 L\zeta^2
\end{aligned}
\tag{24}
$$

where $\mathbb{D} := \|x^{(0,0)-x^*}\|$. Furthermore, when the client learning rate is chosen as

$$\eta = \min\left\{\frac{1}{4L}, \frac{M^{\frac{1}{2}}\mathbb{D}}{\tau^{\frac{1}{2}}T^{\frac{1}{2}}\sigma}, \frac{\mathbb{D}^{\frac{2}{3}}}{\tau^{\frac{2}{3}}T^{\frac{1}{3}}L^{\frac{1}{3}}\sigma^{\frac{2}{3}}}, \frac{\mathbb{D}^{\frac{2}{3}}}{\tau T^{\frac{1}{3}}L^{\frac{1}{3}}\zeta^{\frac{2}{3}}}\right\}, \tag{25}$$

we have

$$
\begin{aligned}
&\mathbb{E}\left[\frac{1}{\tau T}\sum_{t=0}^{T-1}\sum_{k=1}^{\tau} F\left(\overline{\boldsymbol{x}}_t^k\right) - F\left(\boldsymbol{x}^\star\right)\right]\\
&\leq \underbrace{\frac{2L\mathbb{D}^2}{\tau T} + \frac{2\sigma\mathbb{D}}{\sqrt{M\tau T}}}_{\text{Synchronous SGD}} + \underbrace{\frac{5L^{\frac{1}{3}}\sigma^{\frac{2}{3}}\mathbb{D}^{\frac{4}{3}}}{\tau^{\frac{1}{3}}T^{\frac{2}{3}}} + \frac{19L^{\frac{1}{3}}\zeta^{\frac{2}{3}}\mathbb{D}^{\frac{4}{3}}}{T^{\frac{2}{3}}}}_{\text{Add'l errors from local updates \& non-IID data}}
\end{aligned}
\tag{26}
$$

Now we have proved the convergence of distributed learning, we need to prove the convergence of gradient matching using the distributed model.

We note the distributed model as $\theta_t$. For $opt - \text{alg}_\mathcal{S}\left(D\left(g_\mathcal{T}, \mathfrak{g}_S\right), \varsigma_\mathcal{S}, \eta_\mathcal{S}\right)$, the $D\left(g_\mathcal{T}, \mathfrak{g}_S\right)$ is computed as

$$D\left(g_\mathcal{T}, \mathfrak{g}_S\right) = \|g_\mathcal{T} - \mathfrak{g}_S\|^2 \tag{27}$$

We first provide 3 assumptions:

**Assumption 4: Properties of the objective function is L-smooth.** Assume that the objective function $D\left(g_\mathcal{T}, \mathfrak{g}_S\right)$ (we abbreviate the formula as $D(S) = D\left(g_\mathcal{T}, \mathfrak{g}_S\right)$ in the following discussion) is convex and has Lipschitz continuous gradient, that is, there is a constant $L > 0$ such that for all $S_1$ and $S_2$,

$$\|\bigtriangledown D\left(S_1\right) - \bigtriangledown D\left(S_2\right)\| \leq L\|S_1 - S_2\|. \tag{28}$$

**Assumption 5: Learning rate.** The learning rate $\eta_\mathcal{S}$ satisfies

$$0 < \eta_\mathcal{S} < \frac{2}{L}. \tag{29}$$

**Assumption 6: Target Function has lower bound.** Assume that the objective function $D(S)$ has a lower bound $D^*$, that is, for all $S$, there is

$$D(S) \geq D^* \tag{30}$$

Performing a first-order Taylor expansion at $S_{t+1}$ for $D(S)$, we have

$$G(S_{t+1}) \approx G(S_t) + \bigtriangledown G(S_t)^\top(S_{t+1} - S_t). \tag{31}$$

Following update rules based on gradient descent $S_{t+1} = S_t - \eta_{\mathcal{S}} \triangledown D(S_t)$, we can substitute this into the above Taylor expansion and get

$$D(S_{t+1}) \approx D(S_t) - \eta_{\mathcal{S}} || \triangledown D(S_t) ||^2. \tag{32}$$

Since the gradient of $G(S)$ is Lipschitz continuous, we have

$$D(S_{t+1}) \leq D(S_t) + \triangledown D(S_t)^\top (S_{t+1} - St) + \frac{L}{2} ||S_{t+1} - St||^2. \tag{33}$$

Substituting the gradient descent update rule, we have

$$D(S_{t+1}) \leq D(S_t) - \eta_{\mathcal{S}} || \triangledown D(S_t) ||^2 + \frac{L\eta_{\mathcal{S}}^2}{2} || \triangledown D(S_t) ||^2. \tag{34}$$

Simplifying the above inequality, we get

$$D(S_{t+1}) \leq D(S_t) - (\eta_{\mathcal{S}} + \frac{L\eta_{\mathcal{S}}^2}{2}) || \triangledown D(S_t) ||^2. \tag{35}$$

Because $0 < \eta_{\mathcal{S}} < \frac{2}{L}$, so $\eta_{\mathcal{S}} + \frac{L\eta_{\mathcal{S}}^2}{2} > 0$, which suggests that $D(S_{t+1}) \leq D(S_t)$. This indicates that the function value gradually decreases with iteration.

Then starting from $D(S_{t+1}) \leq D(S_t) - \left( \eta_{\mathcal{S}} - \frac{L\eta_{\mathcal{S}}^2}{2} \right) \|\nabla D(S_t)\|^2$, we can accumulate the reductions across all iterations. For any T iterations, we have

$$D(S_T) - D(S_0) \leq -\sum_{t=0}^{T-1} \left( \eta_{\mathcal{S}} - \frac{L\eta_{\mathcal{S}}^2}{2} \right) \|\nabla D(S_t)\|^2 \tag{36}$$

Then we have

$$\sum_{t=0}^{T-1} \|\nabla D(S_t)\|^2 \leq \frac{D(S_0) - D(S_T)}{\eta_{\mathcal{S}} - \frac{L\eta_{\mathcal{S}}^2}{2}} \tag{37}$$

Since $D(S) \geq D^*$, we can substitute the lower bound $D^*$ into the above inequality to get

$$\sum_{t=0}^{T-1} \|\nabla D(S_t)\|^2 \leq \frac{D(S_0) - D^*}{\eta_{\mathcal{S}} - \frac{L\eta_{\mathcal{S}}^2}{2}} \tag{38}$$

The above inequality shows that as the number of iterations $T$ increases, there is an upper bound to the sum of squared gradients. This means that as iterations proceed, the size of the gradients must decrease because their cumulative sum is finite. Therefore, we can infer that $\triangledown D(S_t)$ tends to zero as $t$ increases, which means that $D(S)$ will converge within some bound. This bound is determined by the initial function value $D(S_0)$ and the theoretical minimum value $D^*$.

## B  Privacy Analysis

DISTDD adds DPSGD to protect privacy; here, we give the privacy guarantee for DPSGD in DISTDD.

First, we review the definition of differential privacy. A randomized algorithm $\mathcal{A}$ satisfies $(\epsilon, \delta)$-differential privacy, if for any two adjacent data sets $D$ and $D'$ (they differ in one element), and all $S \subseteq Range(\mathcal{A})$, have:

$$P(\mathcal{A}(D) \in S) \leq e^\epsilon P(\mathcal{A}(D') \in S) + \delta \tag{39}$$

Among them, $e^\epsilon$ represents the upper bound of privacy loss, and $\delta$ represents the probability upper bound that the algorithm may completely violate $\epsilon$-differential privacy.

In DISTDD, DPSGD achieves differential privacy by adding noise during gradient calculation. Specifically, for each training sample, we calculate its gradient, clip it to limit its L2 norm, and add random noise that satisfies the Gaussian distribution. This process can be formalized as:

- Gradient calculation: For each sample $x_i$, calculate the gradient of the loss function $L(\theta, x_i)$ with respect to the model parameters $\theta$ $g_i = \nabla_\theta L(\theta, x_i)$.

- Gradient clipping: clip each gradient $g_i$ to the maximum L2 norm $C$, and get $\tilde{g}_i = g_i / \max(1, \frac{\|g_i\|_2}{C})$.

- Add noise: Calculate the average value of the clipped gradient, and add noise that satisfies the Gaussian distribution $N(0, \sigma^2 C^2 I)$, where $\sigma$ is the standard deviation of the noise, $I$ is the identity matrix. That is, $\hat{g} = \frac{1}{n} \sum_{i=1}^{n} \tilde{g}_i + N(0, \sigma^2 C^2 I)$.

The Gaussian mechanism shows that for any function $f$, if we add Gaussian noise with mean $0$ and standard deviation $\sigma$ to its output, then we can achieve $(\epsilon, \delta)$-differential privacy, where $\epsilon$ and $\delta$ are related to the standard deviation of the noise $\sigma$, function $f$ in any two adjacent data sets The maximum output difference is related to the L2 norm $\Delta f$.

For DPSGD, each gradient is clipped to the maximum L2 norm $C$ before adding noise, so for any two adjacent data sets, the maximum difference in gradients (i.e., $\Delta f$) is limited In $2C$.

According to the theorem of Gaussian differential privacy, for a given $\delta$, $\epsilon$ can be calculated by the following formula:

$$\epsilon = \sqrt{2\ln(1.25/\delta)} \cdot \frac{\Delta f}{\sigma} \tag{40}$$

Substituting $\Delta f = 2C$, we get:

$$\epsilon = \sqrt{2\ln(1.25/\delta)} \cdot \frac{2C}{\sigma} \tag{41}$$

Here, $\sigma$ is the standard deviation of Gaussian noise added to the clipped gradient mean, $C$ is the threshold for gradient clipping, $\delta$ is defined in $(\epsilon, \delta)$-The upper bound on the probability of privacy leakage allowed in differential privacy.

Through the above formula, we can see that $\epsilon$ (privacy loss) and the standard deviation of noise $\sigma$, the gradient clipping threshold $C$ and the allowed privacy leakage probability $\delta$ D. Increasing the standard deviation of noise $\sigma$ can reduce $\epsilon$ and thereby enhance privacy protection, but this may be at the expense of model accuracy. On the contrary, reducing $\sigma$ or $C$ can improve model performance but increase the privacy loss $\epsilon$.

## C  Ablation Study

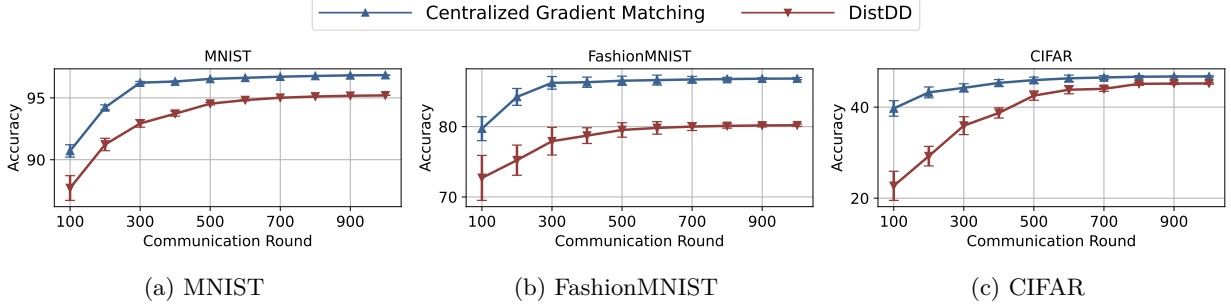

Figure 10: Ablation study of different Communication Rounds.

### C.1  Different Nodes Number

We evaluate the performance of DISTDD in response to varying degrees of node participation. In this particular experiment, it is notable that the cumulative volume of data samples across all clients remains unaltered. Consequently, as we increase the number of participating nodes, the number of data samples

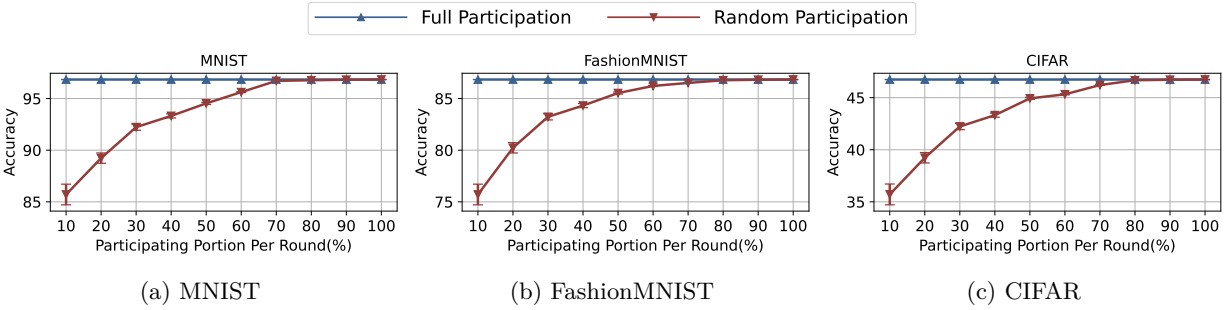

Figure 11: Ablation study of different participants portion.

allocated to each individual client simultaneously diminishes. We rely on the classification accuracy outcomes to illuminate the performance changes, as shown in Figure 12.

To conduct the comparative analysis, we compare three distinct configurations: firstly, the local gradient matching; secondly, DISTDD featuring full participation from all nodes; and thirdly, DISTDD with a 50% random client participation scheme. The experiment results manifest a notable trend. Specifically, the performance of DISTDD with full participation exhibits a gradual decline with the amplification of node numbers; nonetheless, this decline is relatively modest. In contrast, the performance of DISTDD with random participation shows a substantially steeper descent in accuracy.

## C.2 Image number per class

In this section, we explore the impact of the number of generated images per class with a specific focus on its effect on classification accuracy. To undertake this ablation study, we systematically vary the quantity of images per class, encompassing the values 1, 10, 20, 30, 40, and 50. The outcomes are shown in Figure 13.

It is notable that local gradient matching reaches convergence primarily when the image count per class ranges between 10 and 20. In contrast, DISTDD exhibits a convergence behavior at a significantly higher threshold, typically exceeding 30 images per class. This observation suggests that DISTDD necessitates a more substantial quantity of images to aggregate knowledge from the distributed clients effectively. However, it is noteworthy that the performance of DISTDD demonstrates the potential to approximate the performance levels achieved by local gradient matching when the image count per class reaches sufficiently high values. This disparity in the requisite image count for DISTDD may be attributed to the expansive dispersion of data across numerous clients, consequently mandating a greater number of generated images to facilitate convergence.

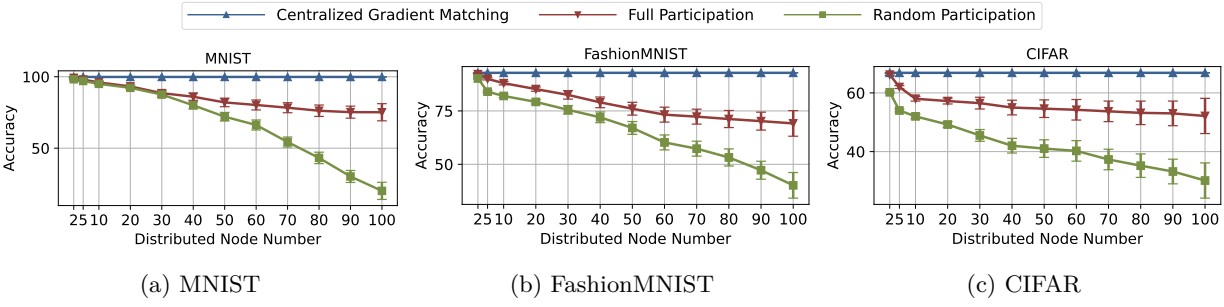

Figure 12: Ablation study of node number.

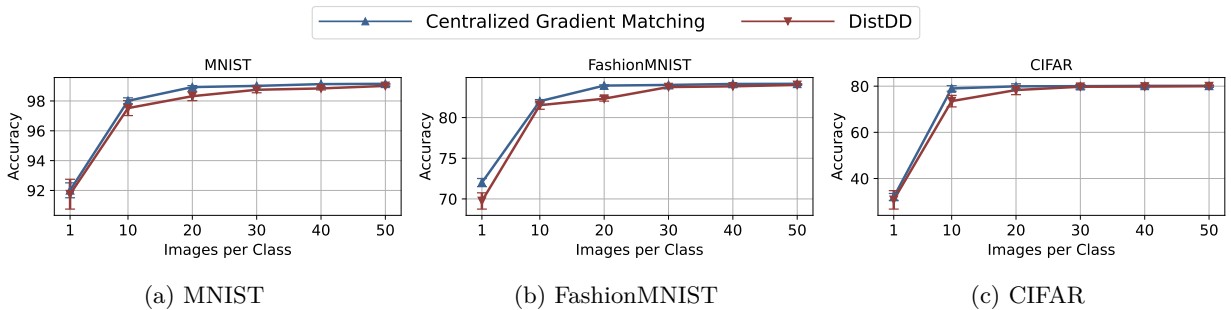

(a) MNIST          (b) FashionMNIST          (c) CIFAR

Figure 13: Ablation study of different image numbers per class.

### C.3 Communication Rounds

In this section, we evaluate the influence of communication rounds on the performance of DISTDD with a particular emphasis on its impact on classification accuracy. We conduct this analysis by contrasting two configurations of DISTDD one with full client participation per round and another with random participation of 50% of the clients per round, within the context of a 20-client scenario. The results, illustrated in Figure 10, offer the observed effects.

Evidently, DISTDD with full client participation typically requires approximately 300 communication rounds to converge. In contrast, the variant of DISTDD featuring random client participation necessitates a significantly greater number of communication rounds to achieve the same convergence. This discrepancy in the convergence rate primarily stems from random client participation, which mandates a more extended communication process for each client to convey and synchronize their knowledge with the central server.

### C.4 Portion of Selected Clients per Round

Next, we study the effect of the proportion of selected clients per round, focusing on random participation throughout 500 communication rounds. We maintain a constant client count of 20 while adhering to a Dirichlet distribution parameter ($dir = 1.0$) for data partitioning. The proportion of participating clients is systematically varied, ranging from 10% to 100% (representing full participation).

Noteworthy is the observation that it necessitates a participation rate of 80% within the random selective participation scheme to achieve parity in classification accuracy with full participation. Conversely, when the participation rate falls below the 50% threshold, the performance of DISTDD markedly falls behind that of local gradient matching. This disparity in performance underlines the significance of the participation proportion in the context of random selection and underscores the trade-off between participation rate and classification accuracy.

## D Adding Differential Privacy Noise to DistDD

Moreover, our investigation extends to assessing the influence of incorporating differential privacy (DP) mechanisms into our DISTDD. Differential privacy has proven its efficacy in protecting individual privacy within distributed learning frameworks, rendering it an appealing avenue for augmenting privacy assurances among participating clients. In the context of this experimental study, we systematically vary the noise scale parameter denoted as $\sigma$, exploring values ranging from 0.01 to 100. This comprises a comparative analysis of DISTDD's performance in the absence of DP (referred to as the non-DP scenario) and its performance when DP is integrated (referred to as the DP-enabled scenario).

As shown in Figure 14, our findings substantiate that when the noise scale $\sigma$ surpasses the threshold of 1e-2, a pronounced detrimental effect on DISTDD's performance becomes evident. Notably, the outcome is manifested as a substantial degradation in the system's overall performance metrics.

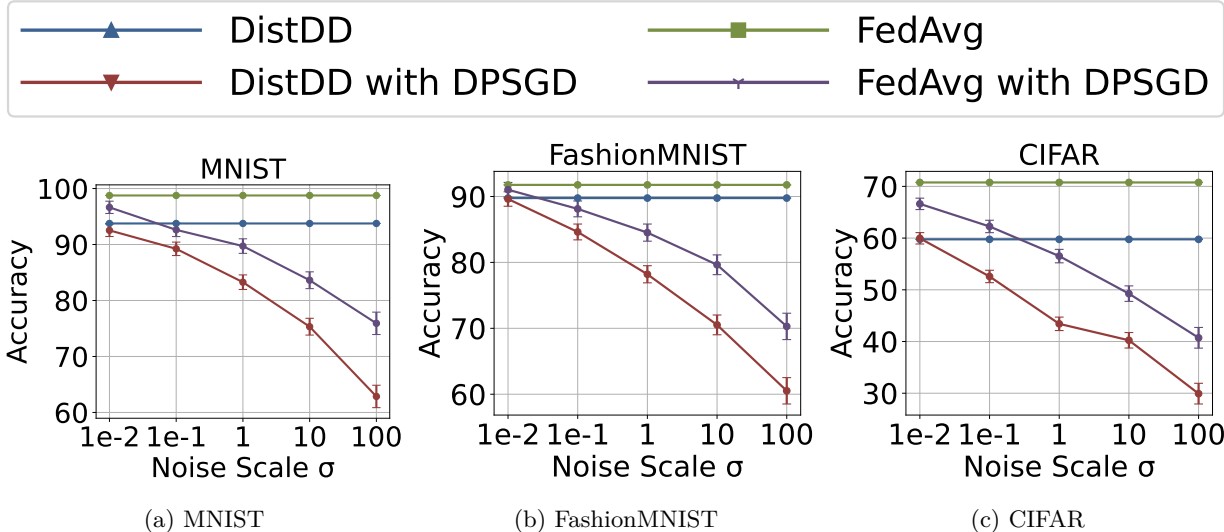

(a) MNIST          (b) FashionMNIST          (c) CIFAR

Figure 14: This study explores the impact of integrating differential privacy (DP) into DISTDD, a system used within distributed learning environments to enhance privacy. By adjusting the noise scale parameter, $\sigma$, from 0.01 to 100, the study compares the performance of DISTDD with and without DP. The findings reveal that increasing $\sigma$ beyond 0.01 significantly diminishes DISTDD's performance, resulting in a marked reduction in its overall efficiency. This indicates that while DP adds a layer of privacy protection, it also poses challenges by adversely affecting system performance when the noise level is too high.

In summary, this comprehensive exploration underscores the critical significance of judiciously configuring the noise scale parameter when integrating differential privacy into DISTDD, thus ensuring that privacy enhancements are harmoniously balanced with the preservation of system performance and convergence integrity.

