# OpenReview forum: "DistDD: Distributed Data Distillation Aggregation through Gradient Matching"
_TMLR — Rejected by TMLR_

### Review · Reviewer_wgwR · 2025-03-20

**Summary Of Contributions:**

In this work, authors propose a novel method called DistDD. It is a federated learning framework with data distillation using gradient matching. This distilled dataset supports potential in-time parameter tuning and NAS without the need of retraining whole FL framework, which can reduce communication costs. A lot of experiments are also conducted to proof the ability of DistDD in non-iid and mislabeled data situation.

**Audience:**

No

**Broader Impact Concerns:**

No Broader Impact Concerns.

**Claims And Evidence:**

No

**Requested Changes:**

1. Some studies have been conducted to use dataset distillation in FL, e.g. FedD3 [1]. What's the difference between your work and other related works for incorporating dataset distillation to FL? Authors didn't have discussion about them in related works and also comparison experiments.
2. Experimental settings is not clear. Author mentioned using CNN as backbone model, but which CNN?
3. In Figure 2, when mislabeling data portion is 0.0, why FedAvg is worse than DistDD? Since authors didn't mention the experiment setting in this part, I suppose the result should be align with Table 1, where fedavg performs better.
4. Figure citation in some sections are inaccurate. E.g., sec 5.1 and 5.2 cited figure 12 and figure 13 which presented in supplemental materials. They should be figure 8 and figure 9 based on context. Also, figure layout is not good. For example, sec 4.4 is discussing non-iid, but the figure there is use case.
5. In the introduction part, there is a sentence 'To reduce such communication costs, an appealing approach is for the clients to upload the data directly to the server so that future training and tuning can only happen within the server.' This statement against the purpose and principle of FL. If you want to discuss data sharing in FL, it is better to introduce some method related to partial data sharing like FedAvg-Share [1].
6. Citation format is not accurate. There's difference between using '\citet' and '\citep'. See TMLR submission guidelines.
7. I will recommend authors to have experiments on accuracy comparison with fedavg and fedprox directly show the advantage of DistDD in reducing communication cost. For example, under very limited communication rounds for these methods, can DistDD outperforms others?

[1] Song, Rui, et al. "Federated learning via decentralized dataset distillation in resource-constrained edge environments." 2023 International Joint Conference on Neural Networks (IJCNN). IEEE, 2023.
[2] Zhao, Yue, et al. "Federated learning with non-iid data." arXiv preprint arXiv:1806.00582 (2018).

**Strengths And Weaknesses:**

**Strength**

1. Propose a novel method of constructing a distilled dataset though gradient matching in federated learning
2. Conducted extensive experiments across various tasks, like non-iid, mislabel, and user case.

**Weakness**
1. The motivation of this paper is unclear. In introduction, authors use several paragraphs to illustrate the specific user case (parameters fine-tuning and NAS) of DistDD. However, there's no explanation on why we need NAS and further parameter tuning after FL training. Why we need parameter tuning, is there new data come in at some clients? In Figure 4, the result of 10 times further tuning of DistDD is even worse than FedAvg without tuning.
2. When sharing data (including modified dataset) in FL, the top concern is data privacy. Authors mentioned sharing gradient will lead to leak information, and used DPSGD to clip the updated gradient. But this part lack of theoretical or experimental explanation on why using this method can effectively protect privacy.
3. The contribution of this paper is limited. In the main result Table 1, proposed DistDD all underperforms fedavg and fedprox, in all dataset and settings.

---

### Review · Reviewer_PJ9U · 2025-03-26

**Summary Of Contributions:**

This paper introduces DistDD (Distributed Data Distillation through Gradient Matching), a novel framework for federated learning (FL) that addresses communication inefficiencies during hyperparameter tuning and neural architecture search (NAS). The main contributions are:
1. DistDD aggregates distributed client data into a synthetic dataset via gradient matching, enabling server-side parameter tuning and NAS without repeated client communication. This reduces communication overhead by decoupling model optimization from iterative FL rounds.
2. Introduces a median-based defense to mitigate errors from mislabeled data, enhancing robustness. Demonstrates convergence under non-iid distributions using Dirichlet sampling, showing resilience in heterogeneous data scenarios.
3. Provides rigorous convergence proofs for both distributed learning and gradient matching processes. Validates performance on MNIST, FashionMNIST, and CIFAR-10, achieving accuracy comparable to FedAvg while reducing communication costs in NAS and hyperparameter tuning use cases.
DistDD offers a privacy-preserving, communication-efficient alternative to traditional FL for iterative optimization tasks, opening new avenues for distributed learning in resource-constrained environments.

**Audience:**

Yes

**Broader Impact Concerns:**

NA.

**Claims And Evidence:**

Yes

**Requested Changes:**

1. Optimize synthetic data generation to improve convergence under low images-per-class (IPC < 100) in non-IID scenarios.
2. Develop hybrid differential privacy mechanisms (e.g., adaptive clipping + noise) to balance privacy and accuracy tradeoffs.
3. Design client selection strategies (e.g., importance sampling) to reduce required participation rate for random client selection.

**Strengths And Weaknesses:**

Strengths
1. DistDD introduces a distributed data distillation approach that reduces communication overhead in FL by generating a synthetic dataset for server-side tuning/NAS, avoiding repetitive client interactions. Addresses critical use cases (parameter tuning, NAS) that are bottlenecks in traditional FL.
2. Integrates a median-based defense to mitigate errors from mislabeled data, outperforming FedAvg under high mislabeling rates.
3. Demonstrates convergence under Dirichlet-distributed non-iid data, showing adaptability to heterogeneous client data.
4. Provides convergence proofs for both distributed learning and gradient matching processes, establishing mathematical stability.
5. Extensive experiments on MNIST, FashionMNIST, and CIFAR-10 validate performance gains and communication savings in NAS/parameter tuning.

Weaknesses
1. DistDD underperforms FedAvg when the number of images per class (IPC) is below 100, particularly in highly non-iid scenarios (dir=0.1).
2. Integrating differential privacy (DP) with Gaussian noise degrades performance significantly for σ > 0.01, highlighting a need for optimized DP mechanisms.
3. Random participation schemes require ≥80% client participation to match full-participation accuracy, limiting scalability in large-scale or dynamic FL settings.
4. Ablation studies show DistDD requires more synthetic samples (≥30 per class) compared to centralized gradient matching, potentially increasing storage/processing costs.
5. While theoretically proven, DistDD’s robustness to extreme non-iid distributions (e.g., dir < 0.1) is not fully explored, leaving open questions for real-world applications.

---

### Review · Reviewer_Gv3c · 2025-04-20

**Summary Of Contributions:**

This paper proposes DistDD, a federated learning method that reduces communication costs via one-time data distillation, producing a global distilled dataset for efficient parameter tuning and NAS without repeated FL iterations. The authors provide both theoretical convergence and experiments results to validate its robustness and practicality.

**Audience:**

Yes

**Claims And Evidence:**

Yes

**Requested Changes:**

See weakness for details.
1) Provide more clear motivation, especially the optimization target.
2) Provide a baseline in this paper's settings.
3) Provide relative contents of figures in the paper.

**Strengths And Weaknesses:**

Strengths:

* The paper is well structured and easy to follow.
* The authors provide clear and rigorous theoretical analysis for the proposed method.

Weaknesses

* The paper mentions using DistDD for "neural architecture search (NAS) over FL," but it is unclear whether the goal is to:
(a) Perform federated NAS (i.e., collaboratively searching for architectures across clients), or
(b) Use NAS locally on the distilled dataset to improve model performance in a federated setting.
Clarifying the target objective and the role of NAS would strengthen the motivation.

* The experiments only evaluate DistDD with tuned parameters, missing comparisons to other approaches in your settings.
* Can the authors explain the meaning and findings from Figure 6? There seems missing contents of some figures. For example Figure 6.

---

### Decision · Action_Editor_q3vB · 2025-05-29

**Recommendation:** Reject

**Comment:**

While reviewers recognized the novelty of the algorithm in the submission, several important experimental details are missing, and no response has been provided. Therefore, I have decided to reject the submission in its current form. However, the authors are encouraged to address the reviewers' comments when revising the manuscript and to submit a major revision to TMLR in the future.

**Audience:**

Yes

**Claims And Evidence:**

No.

Reviewers raised concerns about the details of the experimental evaluation needed to support the effectiveness of the proposed method, but no response has been provided.

**Resubmission Of Major Revision:**

The authors may consider submitting a major revision at a later time.